# Archaeal histone-based chromatin structures regulate transcription elongation rates
Breanna R. Wenck, Robert L. Vickerman, Brett W. Burkhart & Thomas J. Santangelo ✉

Many archaea encode and express histone proteins to compact their genomes. Archaeal and eukaryotic histones share a near-identical fold that permits DNA wrapping through select histone-DNA contacts to generate chromatin-structures that must be traversed by RNA polymerase (RNAP) to generate transcripts. As archaeal histones can spontaneously assemble with a single histone isoform, single-histone chromatin variants provide an idealized platform to detail the impacts of distinct histone-DNA contacts on transcription efficiencies and to detail the role of the conserved cleavage stimulatory factor, Transcription Factor S (TFS), in assisting RNAP through chromatin landscapes. We demonstrate that substitution of histone residues that modify histone-DNA contacts or the three-dimensional chromatin structure result in radically altered transcription elongation rates and pausing patterns. Chromatin-barriers slow and pause RNAP, providing regulatory potential. The modest impacts of TFS on elongation rates through chromatin landscapes is correlated with TFS-dispensability from the archaeon *Thermococcus kodakarensis*. Our results detail the importance of distinct chromatin structures for archaeal gene expression and provide a unique perspective on the evolution of, and regulatory strategies imposed by, eukaryotic chromatin.

Most Archaea and all Eukarya encode histone proteins that bind DNA to form dynamic chromatin landscapes that compact and organize the genome, thereby impacting transcription and gene expression[1–7]. A few bacterial clades also encode histone-fold containing proteins[8] that interact with DNA very differently from the histone-DNA interactions preserved in archaeal and eukaryotic systems. When sufficiently abundant, archaeal histones spontaneously oligomerize to generate extended archaeal histone-bound chromatin structures that organizes the genome and regulates the progression of the transcription apparatus[6,9]. Archaeal RNA polymerase (RNAP) and eukaryotic RNA polymerase II (Pol II) are structurally and functionally homologous and both must overcome nearly identical histone-bound DNA barriers[10–12]. Chromatin architecture can provide regulatory potential during transcription elongation, alter positions of transcription pausing, and is known to impact elongation-termination decisions[13,14]. Much of what is understood about how histone-based chromatin modulates the transcription apparatus comes from studies targeting eukaryotic histone post-translational modifications (PTMs), epigenetic markers, and chromatin remodeling complexes, all of which influence gene expression[15–17].

Archaea are the likely progenitors of Eukarya[18–23] and the core histone-fold and DNA binding activities of archaeal histones are shared with their eukaryotic counterparts[4,5,24–28]. Archaeal histones retain the canonical histone fold of three alpha helices joined by two loops (α1-L1-α2-L2-α3). Archaeal histones can form both homo- and hetero-dimers that protect ~30 bp of DNA and assemble into an extended, continuous super-helical structure. The geometry of the DNA bound within an archaeal chromatin superhelix nearly exactly matches that of the eukaryotic nucleosomal DNA arrangement[4,24,29–32] and the overall archaeal histone-based extended chromatin structure closely matches chromatin structures found on eukaryotic telomeres[33]. Both archaeal and eukaryotic histone-DNA interactions align to the same nucleosome positioning code and the specific protein-DNA contacts that stabilize chromatin are conserved[29,34–36]. Archaeal genomes, however, appear devoid of chromatin remodeling complexes. Additionally, archaeal histones typically lack the canonical N- and C-terminal extensions common to their eukaryotic counterparts, and PTMs of archaeal histones have not been demonstrated to be either abundant or biologically relevant[37–39].

Although a single archaeal histone isoform is sufficient to spontaneously form extended chromatin structures in vitro, Archaea that encode histone proteins often encode and can differentially express multiple histone isoforms[19,40,41]. It is possible that chromatin assembled from different

Department of Biochemistry and Molecular Biology, Colorado State University, Fort Collins, CO 80523, USA. ✉e-mail: thomas.santangelo@colostate.edu

archaeal histone isoforms adopts unique structures that differentially impact transcription and genome organization[41]. The model archaeal species *Thermococcus kodakarensis* encodes two histone proteins – HTkA and HTkB – that are individually non-essential; deletion of both HTkA and HTkB is synthetically lethal, indicating histone-based chromatin is essential for informational processing from, and replication of, DNA[39]. Strains lacking HTkA or HTkB can be modified at the sole remaining histone-encoding locus to generate strains with single-histone variant chromatin structures. Substitution of even a single histone residue can radically increase or decrease DNA affinity, disrupt the three-dimensional (3D) structure of archaeal chromatin, or disrupt dimer interactions with dramatic impacts to gene expression, growth rates, and overall fitness[4,29,42]. Outstanding questions remain regarding how the archaeal histone-based chromatin landscape impacts gene expression, what roles conserved transcription factors play in assisting the archaeal RNAP when transcribing histone-based chromatin, and how chromatin organization patterns impact the rate of RNA synthesis and pausing for archaeal RNAP.

Archaeal transcription systems are component-simplified but homologous to their eukaryotic counterparts[43-47]. Recapitulation of the archaeal transcription system in vitro using histone-bound templates provides an ideal and complementary platform to delineate the regulation, pausing, and elongation rates of archaeal transcription through varied chromatin landscapes.

To detail the role specific individual residues within the archaeal histone-DNA complex have on the progression of the transcription apparatus, we describe in vitro RNAP processivity in histone-free, histone-bound, and variant histone-bound environments. Individual histone variants, once assembled into archaeal histone-based chromatin, can elicit dramatic changes in the rate of RNA synthesis and pausing patterns during elongation that resolve the roles of select histone-histone and histone-DNA interactions on transcription elongation.

Addition of the well-conserved elongation factor Transcription Factor S (TFS) mildly assists elongation rates and increases full-length RNA production, but TFS activity has negligible effects on RNAP processivity in specific chromatin landscapes designed to disrupt the 3D structure of, or stabilize histone-DNA interactions within, archaeal histone-based chromatin. The minimal impacts of TFS in vitro adumbrated that TFS-activities may not be essential in vivo. The successful, yet phenotypically limited deletion of TFS (TK0533) from *T. kodakarensis*, suggests that backtracking and rescue of archaeal ternary elongation complexes (TECs) in a histone-based chromatin environment, via TFS-stimulated endonucleolytic cleavage by RNAP of nascent transcripts, is not a critical component of archaeal transcription regulatory mechanisms.

Our results reveal how changes to specific histone residues alter chromatin structures that regulate transcription elongation rates and pausing patterns. Based on our results, differential expression and assembly of archaeal histone isoforms could be employed as a regulatory mechanism to control gene expression and genome accessibility. Finally, the minimal impact resultant from deletion of TFS, one of only a few well-conserved archaeal factors known to influence post-initiation regulation of the archaeal RNAP, implies that transcription backtracking does not impart crucial regulation in vivo in optimal conditions. The congruence of archaeal and eukaryotic chromatin structures permits extrapolation of our results beyond archaeal systems to detail how the evolution of eukaryotic histone isoforms changed the chromatin landscape and likely led to the requirement for chromatin remodeling and histone modifications in Eukarya.

## Results
### Select histone-DNA contacts dramatically alter transcription rates and pausing patterns

Translocation and RNA synthesis by archaeal RNAP is slowed by histone-bound DNA but ternary elongation complex (TEC: RNAP, a DNA template, and nascent RNA) stability is not disrupted by chromatin landscapes[9,48]. The ease of spontaneous assembly of chromatin landscapes with single archaeal histone-isoforms permits evaluation of how variant chromatin landscapes regulate transcription elongation rates and pausing patterns.

To examine the impacts of select histone variants and their associated chromatin landscapes on the archaeal transcription apparatus in vitro, we recombinantly expressed and purified the well-studied *T. kodakarensis* histone A protein (HTkA$^{WT}$)[4,7,19,42,49] and select HTkA$^{variants}$ (Supplementary Fig. 1a). Histones with substitutions at residues known to appreciably increase (HTkA$^{E19K}$, HTkA$^{G52K}$, and HTkA$^{E19K/G52K}$) or decrease (HTkA$^{R20S}$ and HTkA$^{T55L}$) affinity to DNA[29,50], with a substitution known to interfere with the 3D structure of the archaeal superhelix (HTkA$^{G17D}$)[4,42], and with substitutions known to disrupt dimer interactions (HTkA$^{E3A}$, HTkA$^{R11A}$, and HTkA$^{E34A}$)[4,31] were prepared to evaluate and compare the impact of specific histone variants on transcription elongation kinetics and pausing. HTkA normally functions at 85-95 °C and preparations of recombinant HTkA often retained dimeric-interactions even after extensive heating and SDS-PAGE (Supplementary Fig. 1b). Western blotting with anti-HTkA antibodies confirm that the higher order complexes resolved in SDS-PAGE are oligomerized HTkA complexes (Supplementary Fig. 1c).

To determine how histone-variant chromatin landscapes impact TEC activities and pausing patterns, we exploited purified HTkA$^{WT}$ and HTkA$^{variants}$, basal regulatory archaeal transcription components, archaeal RNAP, and our capacity to monitor the elongation patterns of TECs in vitro (Fig. 1). To ensure addition of archaeal histones and resultant chromatin structures did not impede transcription initiation, stalled TECs were first formed on histone-free DNA via initiation at a C-less cassette with only ATP, UTP, and GTP (Fig. 1a). Elongation limited by the absence of CTP generated TECs with +58 nucleotide nascent transcripts (TECs$_{+58}$) that were captured and washed to remove excess rNTPs, including radiolabeled UTP, thereby ensuring that the specific activity of all transcripts >+58 nts were identical. Templates containing TECs$_{+58}$ were then saturated with HTkA$^{WT}$ or HTkA$^{variant}$ proteins to form a chromatin landscape that TECs must traverse to extend nascent transcripts upon elongation restart[6,9] (Fig. 1b). DNA templates included a tandem, 60 base pair (bp) SELEX-derived histone positioning sequence (HPS)[50,51], optimized to bind histones downstream the stalled TECs$_{+58}$ (Fig. 1a).

Upon rNTP addition, elongation rapidly restarts but quickly becomes asynchronous (Fig. 1c). Monitoring the changes in nascent RNA length over time permits evaluation of the ensemble average activities of archaeal TECs as they navigate both histone-free and histone-bound DNA (Fig. 2, Supplementary Fig. 2). By binning the percentage of transcripts according to length (seven-bins; Fig. 2a) and monitoring changes to transcript distribution over time, the totality of TEC elongation kinetics could be visualized (Fig. 2a, b). Elongation on histone-free templates (Fig. 1c, lanes 2–6) reveal several short-lived sequence-specific pause positions during elongation from +58 to +231. Full-length transcripts are evident after even just 15 s and continue to accumulate as more TECs reach the end of the linear template. No individual template position results in substantial TEC pausing, and the accumulation of full-length transcripts plateaus within two minutes following elongation restart (Fig. 2a, b).

To determine the average RNAP elongation rate on histone-free, histone-bound, and variant histone-bound chromatin landscapes in nucleotides/second (nt/s) we calculated the sum of the average density of RNAs in each bin over time. The mean (nt/s) across all timepoints defines how quickly RNAP can traverse the template in each chromatin landscape relative to HTkA-free conditions (Fig. 2c, Supplementary Data 1). The comparison of the average TEC progression between histone-free and histone-bound templates provides important insights into the impacts of distinct archaeal histone-based chromatin structures on elongation rates and pausing patterns (Fig. 2c).

Addition of HTkA$^{WT}$ permits formation of chromatin structures on the DNA template downstream of the stalled TECs$_{+58}$. Upon rNTP restart, a native HTkA-based chromatin landscape results in only modest changes to the pausing patterns and progression of TECs (Fig. 1c, lanes 7–11 & Fig. 2). Pause positions noted on histone-free DNA are mildly accentuated, and a few additional, albeit short-lived, new pause positions emerge that collectively reduce the ensemble rate of transcription elongation by ~20% (Fig. 2c, Supplementary Fig. 2). Formation of chromatin

https://doi.org/10.1038/s42003-024-05928-w                                                                **Article**

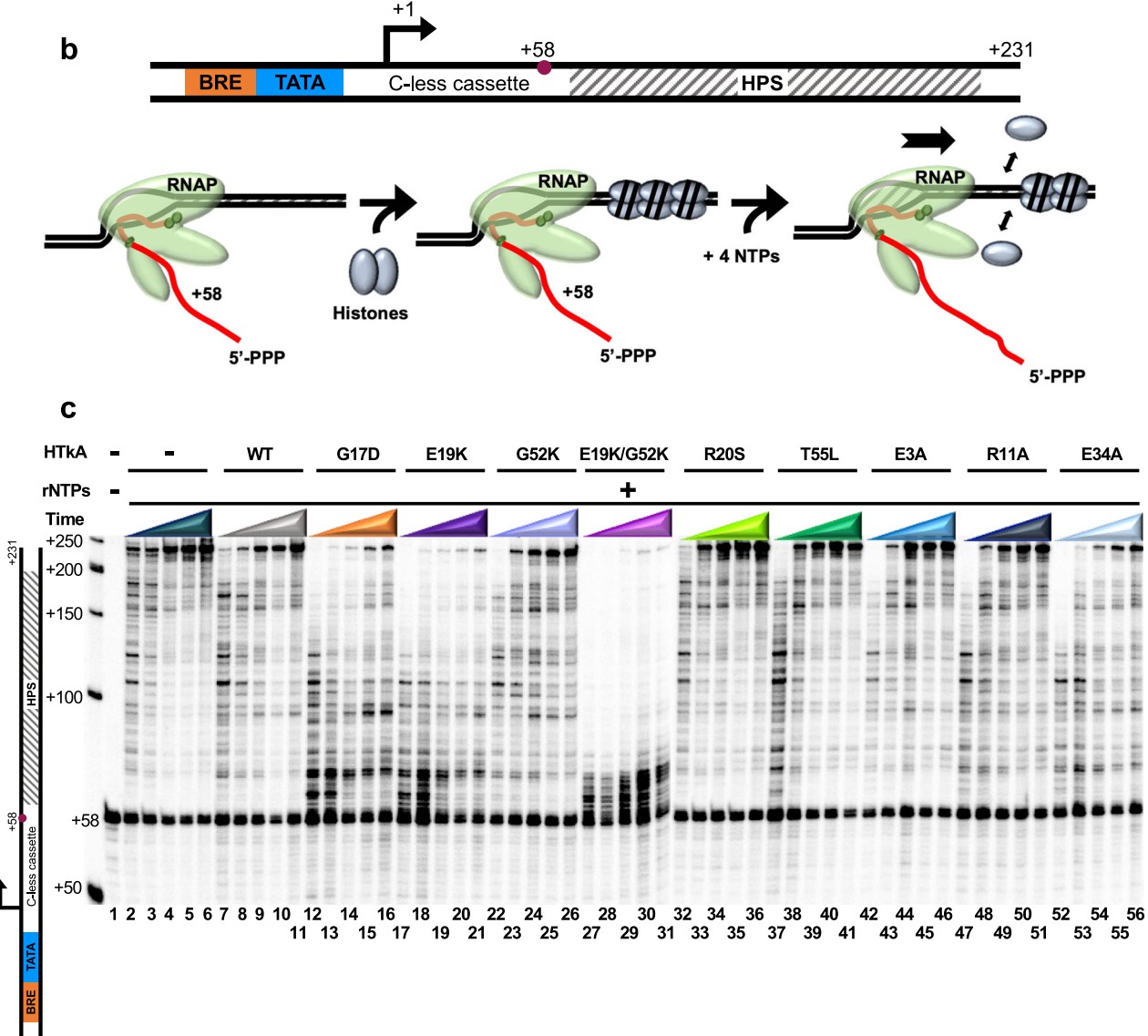

**Fig. 1 | Variant archaeal histone-based chromatin landscapes dramatically alter the rate of RNA synthesis and pausing patterns of transcription elongation complexes. a** DNA template designed with the robust *hmtB* promoter sequence (*italics*), a defined transcription start-site (**bold**), a C-less cassette for a + 58 walk-out (*italics*), and a SELEX-derived double 60 bp histone positioning sequence (HPS) optimized for histone binding. **b** RNA synthesis to +231 was monitored following elongation restart of TECs$_{+58}$ on linear templates that were bound by archaeal histones after TEC$_{+58}$ formation. Addition of RNAP; basal transcription factors (TFB and TBP); and rATP, rGTP, rUTP, and rUT$^{32}$P nucleotides permits synthesis of a body-labeled RNA. To prevent interference of histone-binding with

transcription initiation and to isolate transcription elongation activity, the ternary elongation complex (TEC) is stalled at +58 bp via nucleotide deprivation (rCTP) and the presence of dGTP in the DNA template. Histone proteins are then added to chromatinize the DNA template, followed by addition of all four rNTPs.
**c** Continued RNA synthesis from TECs$_{+58}$ (lane 1) was monitored by revealing changes in nascent transcript length in reaction aliquots removed after 15-, 30-, 60-, 120-, and 240-s following transcription restart upon rNTP addition. Elongation was permitted along DNA lacking any bound proteins (HTkA-free), or histone-bound templates formed with HTkA$^{WT}$ or HTkA$^{variants}$, $n = 4$. Radiolabeled ssDNA makers provide size standards.

Communications Biology| (2024)7:236                                                                                           3

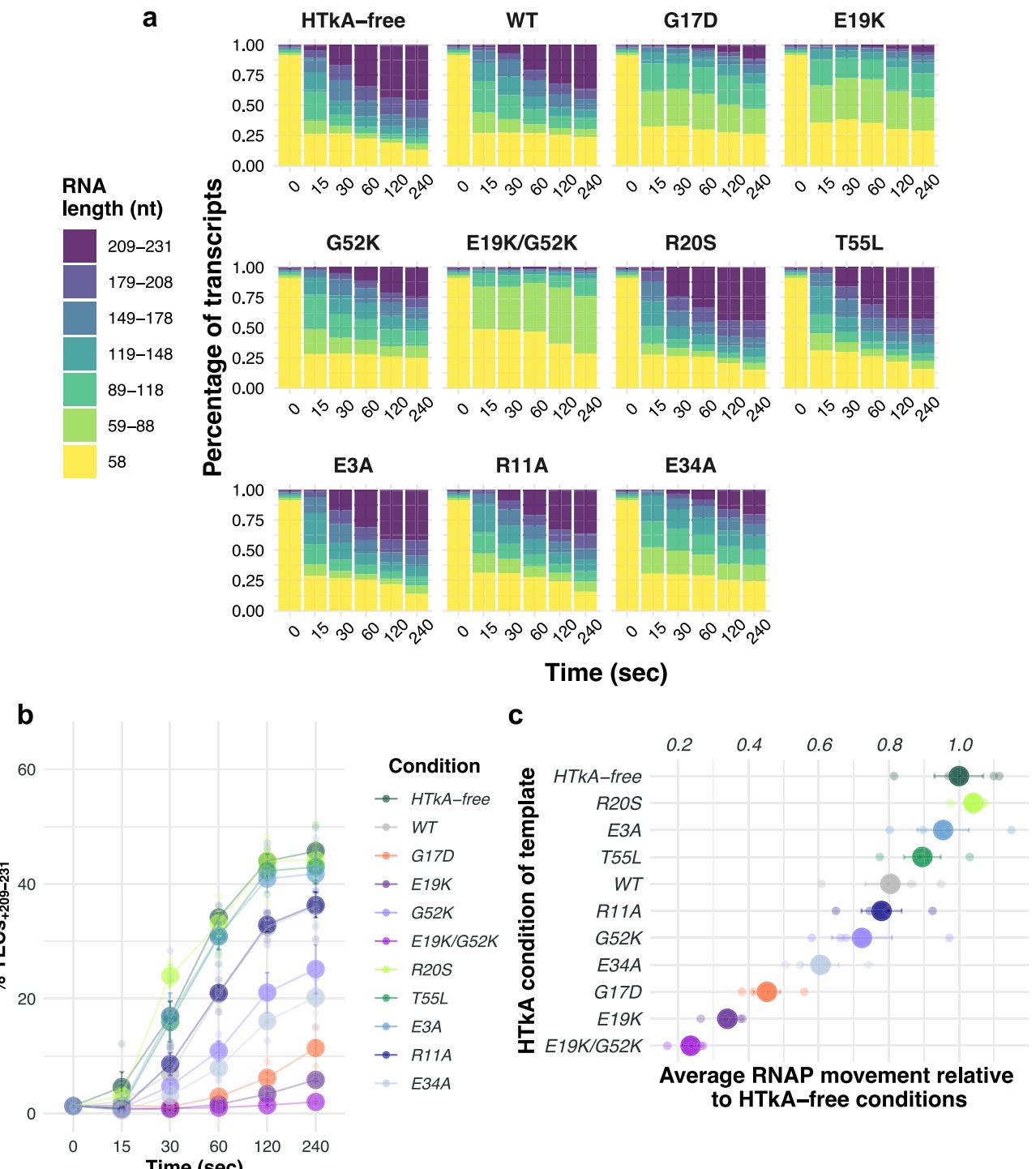

**Fig. 2 | Substitution of key archaeal histone residues has pronounced effects on RNA synthesis and RNAP elongation rates. a** Stacked bar plots quantify the changing percentage of differing transcript lengths (y-axis), divided into seven distinct bins as detailed in the key (left) over time (x-axis), during elongation on non-, WT-, and variant-chromatinized templates. **b** Histone residue substitutions have dramatic effects on full-length RNA transcript abundance. The percentage of RNA transcripts that progress to full-length varies between HTkA-free, HTkA$^{WT}$, and HTkA$^{variant}$ environments. **c** Relative RNAP elongation rates are plotted with respect to the rate of synthesis (nt/s) on templates lacking histone proteins (average elongation rates). Error bars represent the SE from $n = 4$ experiments.

structures downstream of TECs$_{+58}$ has minimal impacts on the percentage of TECs that restart elongation upon rNTP addition. Therefore, isolated archaeal RNAP, without the aid of transcription factors or remodeling complexes, is proficient at elongation on histone-bound DNA in vitro. The minor impacts to elongation rates due to downstream histone barriers likely helps explain the absence of any known chromatin remodeling complexes within archaeal genomes, as archaeal histone-

based chromatin (in WT form) does not dramatically impact TEC translocation and RNA synthesis.

While histone-based chromatin formed with HTkA$^{WT}$ has modest impacts on elongation kinetics and pausing, changing select residues that impact histone-histone or histone-DNA interactions, or that alter the 3D structure of archaeal histone-based chromatin can elicit notable changes to elongation rates and pausing positions (Figs. 1c and 2, Supplementary

Fig. 2). For example, T55 is positioned appropriately to make a salt bridge with R20 from an adjacent monomer[29]. Introduction of histone variants R20S (Fig. 1c, lanes 32–36 & Supplementary Fig. 2f) and T55L (Fig. 1c, lanes 37–41 & Supplementary Fig. 2g) decrease DNA affinity[29,52] and result in minimal elongation conflicts; elongation rates on chromatin formed with HTkA[R20S] or HTkA[T55L] are decreased just ~5–10% from histone-free conditions and thus have less impact on RNAP progression than observed with HTkA[WT]-derived chromatin (Fig. 2). In stark contrast, introduction of histone variants E19K (Fig. 1c, lanes 17–21 & Supplementary Fig. 2c), G52K (Fig. 1c, lanes 22–26 & Supplementary Fig. 2d), or a variant with both E19K/G52K (Fig. 1c, lanes 27–31 & Supplementary Fig. 2e) that appreciably increase DNA affinity[29] result in formation of a chromatin landscape that slows (each single mutant) and nearly impedes (double mutant) transcription elongation upon rNTP restart. The increased interactions between the phosphate backbone of DNA and the positively charged lysine residue(s) effectively impairs TEC movement, reducing elongation rates by ~25% (HTkA[G52K]), ~65% (HTkA[E19K]), and ~75% (HTkA[E19K/G52K]) (Fig. 2c, Supplementary Data 1), nearly eliminating full-length transcript production over the initial time course in the HTkA[E19K/G52K]-based environment (Fig. 2a, b, Supplementary Fig. 2e). Extending the time course of transcription elongation on histone-free templates or through a HTkA[WT]-based chromatin landscape (Fig. 3a, lanes 2–8 and lanes 9–15, respectively) demonstrates that nearly all TECs$_{+58}$ eventually restart elongation upon rNTP addition and that no prominent pause positions dominate elongation rates (Fig. 3a–c). While HTkA[E19K/G52K]-based chromatin landscapes do result in a near complete capture of TECs in an extended pause at ~+70 nts (Fig. 3d), essentially all TECs are eventually capable of independently clearing this pause and elongating towards the end of the template (Fig. 3a, lanes 16–22, b, c).

For HTkA[E19K] and HTkA[E19K/G52K]-based chromatin, TECs encounter a long-lived pause at ~+70 nts, corresponding to the position at which the leading edge of the TEC is likely to collide with a well-positioned and tightly bound histone-dimer at the HPS (Fig. 3d). As the footprint of archaeal RNAP is ~20 bp[53], forward translocation following elongation restart from +58 on our DNA template puts the 3' end of the nascent RNA in the active site of RNAP at ~70 nts if the TEC is subject to pause near the beginning of the HPS (Fig. 1a). Release from the ~+70 nts pause is rate limiting for full-length transcript production (Fig. 3d). Given that histone dimers bind and protect just 30 bp, a nearly identical barrier should be encountered again as TECs reach ~100, 130, 160, and 190 nts, but only the first barrier represents a substantial pause position. The first histone-based chromatin barrier to continued elongation is thus the prominent position of regulation for continued transcription. It is possible that disrupting the chromatin barrier from the most TEC proximal histone-dimer results in changes to the extended chromatin structure that clear the template of major barriers to continued elongation.

The prominent pause at ~+70 nts on templates containing HTkA[E19K/G52K]-based chromatin implies that TECs must wait for downstream histone-dimers to spontaneously release from the template to permit continued elongation (Fig. 3d). Stable HTkA[E19K/G52K]-histone-DNA complexes are confirmed by monitoring elongation for extended times (Fig. 3e–g). After 2 min of elongation, most TECs are still paused at ~+70 nts on HTkA[E19K/G52K]-histone bound templates (Fig. 3e, lanes 2–5). Dilution of the reactions to reduce total histone concentrations (Fig. 3e, lanes 6–8, g), or addition of HTkA[WT] to promote exchange of DNA-bound histones (Fig. 3e, lanes 9–11, g) was not successful in altering elongation rates in comparison to maintaining the identical landscape through addition of HTkA[E19K/G52K] (Fig. 3e, lanes 12–14, g). The results obtained imply that the rates of spontaneous histone dissociation from DNA to permit continued elongation of TECs is slow relative to the rate of transcription elongation.

As observed for histone-variants that increase DNA affinity, introduction of a histone variant that disrupts the 3D structure of chromatin[4] (HTkA[G17D]; Fig. 1c, lanes 12–16 & Supplementary Fig. 2b) results in impaired TEC progression, substantial pausing, and reduced RNA synthesis rates. Given that G17 does not directly contact DNA, it is perhaps surprising that the elongation kinetics do not match HTkA[WT]; HTkA[G17D]-based chromatin reduces elongation rates by ~55% (Fig. 2). A prominent pause site at ~+70 nts demonstrates that TECs still encounter the most TEC-proximal dimer as a major barrier of HTkA[G17D]-based chromatin that is rate limiting for production of full-length transcripts (Fig. 3d). It will thus be critical to evaluate systems wherein histone-isoforms that may impair continued polymer formation are introduced into heterologous archaeal histone-based chromatin structures. Although neither HTkA nor HTkB is predicted to impair polymerization, some histone-isoforms in less genetically tractable systems lacking in vitro transcription systems may provide a mechanism to cap the growth of extended histone polymers[54].

Archaeal histone dimers can themselves dimerize to form tetramers, and polymerization of histone dimers can continue, in theory, indefinitely to form very long extended chromatin structures. While some residues such as T55 and R20 from adjacent monomers are responsible for essential salt bridge formation and DNA interactions (Fig. 4a), some dimer-dimer interactions are crucial for normal elongation rates in a histone-based chromatin environment (Fig. 2c). The dimer-dimer interface is, in part, coordinated by E3, R11, and E34 (Fig. 4b). E34 is situated within the histone-DNA complex to position R11 favorably with E3, allowing a salt bridge to stabilize histone-dimerization[1,4,5,55] (Fig. 4b). Chromatin generated from HTkA[E3A] (Fig. 1c, lanes 42–46 & Supplementary Fig. 2h) only mildly reduces transcription elongation rates when compared to histone-free conditions and permits faster elongation than seen with HTkA[WT]-chromatin (Fig. 2, Supplementary Data 1). While HTkA[E3A]-based chromatin does not result in any new pausing patterns compared to HTkA[WT], chromatin landscapes formed by HTkA[R11A]- or HTkA[E34A]-histone variants do mildly and reasonably hinder RNA synthesis, respectively, with pausing patterns at similar positions as those observed with HTkA[WT], but each with increased duration (Fig. 1c, Supplementary Fig. 2i, j). The pausing is more notable for HTkA[E34A]-based chromatin landscapes that reduce elongation rates to just half that of histone-free conditions (Fig. 2, Supplementary Data 1). Although HTkA[R11A]- and HTkA[E34A]-based chromatin structures increase pausing and decrease elongation rates, neither displays the prominent ~+70 nt pause (Fig. 3d) that controls overall synthesis rates for HTkA[E19K]-, HTkA[E19K/G52K]-, and HTkA[G17D]-based chromatin landscapes.

## Transcription Factor S (TFS) increases productive elongation through chromatin

Nearly all Archaea, including those species that do not encode histone proteins, encode Spt4, Spt5, and Transcription Factor S (TFS)[12]. Spt4/5 and TFS are known transcription factors that directly bind the archaeal RNAP and can accelerate transcription through chromatin landscapes[6]. Near-universal retention of each factor in archaeal genomes implies their importance to transcription regulation and fidelity in vivo. TFS is known to stimulate the intrinsic endonucleolytic cleavage activities of archaeal RNAP, providing a mechanism to escape pauses that result in retrograde movement (i.e., backtracking) of archaeal TECs that encounter barriers to continued forward translocation[6,56,57]. Collisions between the archaeal TEC and chromatin-barriers are expected to result in some backtracking; however, TFS addition to in vitro transcription reactions has little effect on the average RNAP elongation rate from TECs$_{+58}$ that resume elongation upon rNTP addition (Fig. 5, Table 1, Supplementary Data 1). While the addition of TFS has negligible impacts on the average TEC elongation rate through HTkA[WT]-chromatin landscapes in vitro, TFS addition does result in an increase of full-length transcripts by ~10%, largely due to release of a substantial percentage of +58 complexes into active elongation (Fig. 5a–c, Table 1, Supplementary Fig. 4, Supplementary Data 1).

TFS addition to TECs elongating through chromatin landscapes formed with histone-variants produced mixed results (Figs. 2a and 5, Supplementary Fig. 3, Supplementary Data 1). In every case, addition of TFS has a slight prohibitory effect on TEC progression in early timepoints (Fig. 5a, Supplementary Fig. 3), but release from +58 was increased (Table 1, Supplementary Fig. 4). During elongation through HTkA[WT]- and HTkA[variant]-based chromatin landscapes that only modestly impact elongation rates and do not

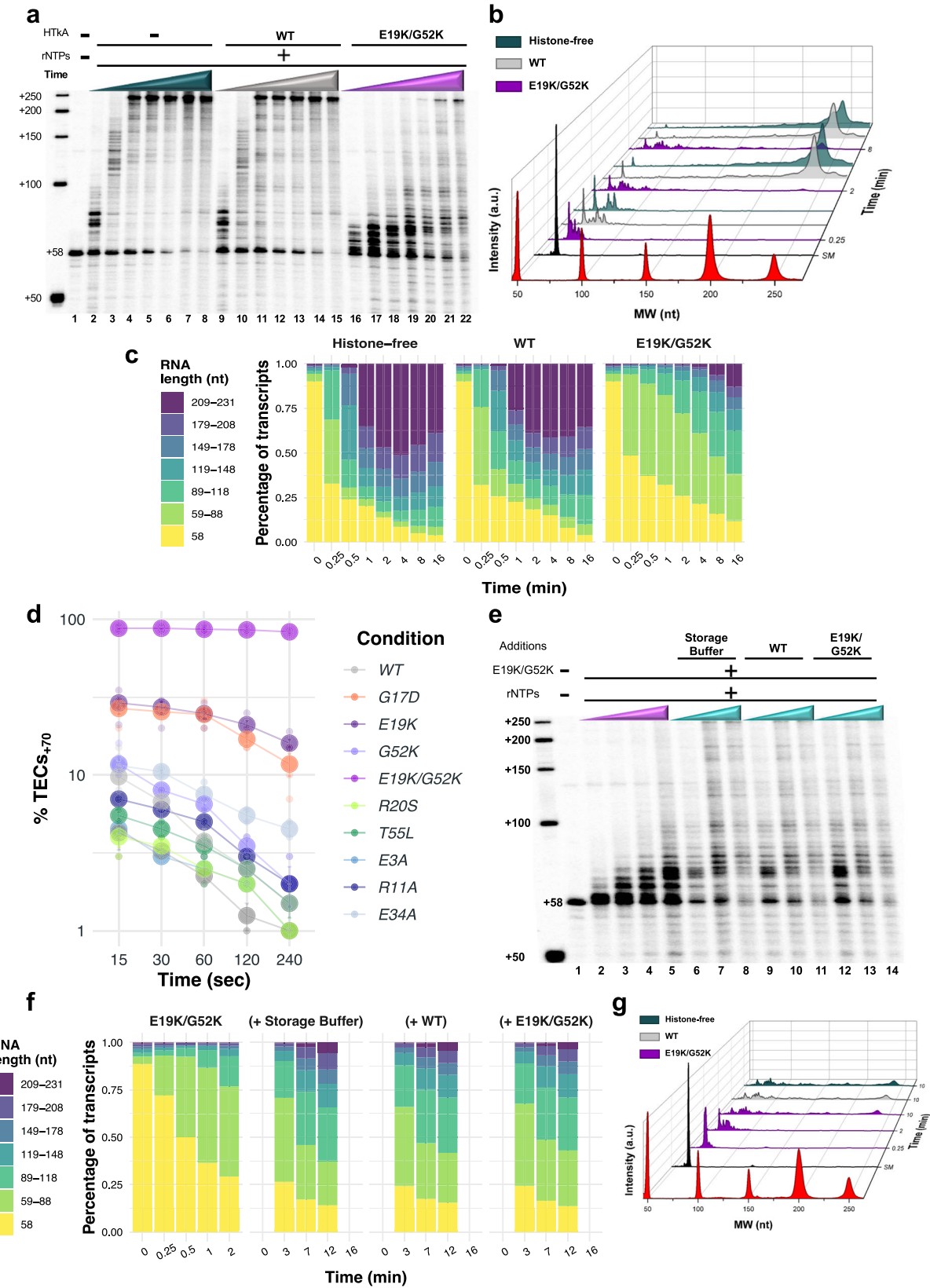

result in a marked percentage of TECs pausing at ~+70 nts, TFS addition has limited influence on the average RNA synthesis rates but results in an increase of full-length RNA products compared to the reactions without TFS (Figs. 2a, b and 5b, c; Supplementary Fig. 3; Supplementary Data 1). Akin to the previous chromatin landscapes, TFS addition to transcription reactions traversing the HTkA$^{E19K}$-, HTkA$^{G17D}$-, or HTkA$^{E19K/G52K}$-based chromatin landscapes had minimal impacts on the average RNA synthesis rate (Figs. 2a, c and 5b, d; Supplementary Fig. 3; Supplementary Data 1). TFS addition was impactful at reducing the +70 nts prominent pause on templates with HTkA$^{E19K}$- or HTkA$^{G17D}$-based chromatin structures

**Fig. 3 | The first point of contact between TECs and downstream chromatin structures is rate limiting for RNA synthesis. a** The dynamics of histone exchange on DNA can dramatically alter TEC progression. Stalled TECs$_{+58}$ were incubated at 85 °C without HTkA (lanes 2−8), with HTkA$^{WT}$ (lanes 9−15) or with HTkA$^{E19K/G52K}$ (lanes 16−22) prior to elongation restart. While elongation on histone-free or HTkA$^{WT}$-bound templates permits rapid accumulation of full-length transcripts, HTkA$^{E19K/G52K}$ bound DNAs require extended incubation to permit RNAP to overcome the chromatin landscape. Radiolabeled ssDNA makers provide size standards. **b** Waterfall plots permit quantification of the distribution of nascent transcript lengths over time; the relative intensity of different transcript lengths was normalized to the sum of the counts in the starting material (SM) within each lane. **c** Transcript distributions are quantified to reveal the impediment to elongation imposed by HTkA$^{E19K/G52K}$ bound DNAs. **d** While TECs can traverse a WT

chromatin landscape with minimal pausing, increasing the strength of histone-DNA contacts or disrupting the L1-L1 interface of archaeal histone-based chromatin transiently pauses the majority of TECs at +70 nts. **e** Dynamic exchange of histones on DNA is slow compared to TEC translocation. Stalled TECs$_{+58}$ were incubated with HTkA$^{E19K/G52K}$, then transcription was reinitiated by addition of all 4 rNTPs, and aliquots were collected after 0.25 and 2 min to monitor RNA synthesis. TECs remaining after 2 min were split into thirds, and either storage buffer, HTkA$^{WT}$, or additional HTkA$^{E19K/G52K}$ was added, before allowing additional time (3, 7, and 12 min) for continued RNA synthesis. Radiolabeled ssDNA makers provide size standards. **f, g** Quantitative comparisons of transcript lengths following dilution, exchange with HTkA$^{WT}$, or maintenance of HTkA$^{E19K/G52K}$- based chromatin reveal negligible difference in TEC progression, implying that histone exchange is limited during the time course of TEC translocation through the chromatin landscape.

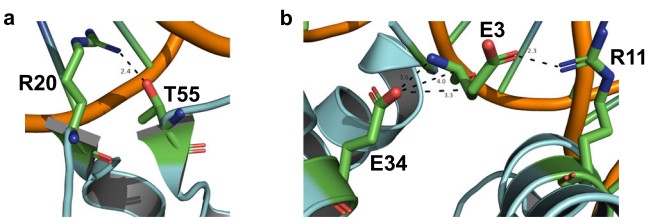

**Fig. 4 | Individual residue substitutions elicit large impacts on the stability of archaeal histone-based chromatin landscapes.** Histone variants with decreased DNA affinity and altered dimer interactions form chromatin structures that minimally impact transcription elongation. **a** T55 and R20 from an adjacent monomer form an intermolecular salt bridge. Disrupting this salt bridge decreases histone-DNA affinity, permitting increased RNA synthesis rates. **b** Histone dimer interactions are facilitated by E34 interactions with R11 and E3.

(Figs. 3d and 5c), implying that cleavage-stimulatory activities of TFS rescued some backtracked TECs and permitted more rapid release from the +70 nts prominent pause. However, TFS-mediated activity was not sufficient to reduce the prominent and rate-limiting pause at ~+70 nts of the HTkA$^{E19K/G52K}$-based chromatin structure (Figs. 3d and 5c).

## TFS (TK0533) is dispensable with relatively minor impacts on overall fitness

The impact of TFS on the percentage of full-length products through histone-based chromatin landscapes are often positive, but the prominent pauses that delay continued elongation on some templates do not behave as would be expected from TECs backtracking due to collisions with chromatin barriers. The absence of TFS from some archaeal genomes, coupled with the meager impacts on transcription rates in vitro, led to attempts to generate a strain of *T. kodakarensis* wherein TK0533 (encoding TFS) was deleted. While previous attempts suggested TFS may be an essential protein[6], continued genetic efforts were successful in deleting TK0533, resulting in strain RLV2 (Supplementary Fig. 5). Markerless deletion[58] of the full sequences encoding TFS was first confirmed through diagnostic PCRs using DNA purified from strain RLV2. The exact endpoints of the TK0533 deletion, and the absence of any second site mutations throughout the entire 2.08 Mbp genomes, were confirmed via whole genome sequencing (WGS) with over 100X coverage (Supplementary Fig. 5a). Despite the positive effects of TFS on backtracked TECs traversing some chromatin landscapes in vitro, the absence of TFS does not result in considerable growth defects (Supplementary Fig. 5b), implying that the activities of the archaeal RNAP, alone or in combination with Spt4/5, suffice to permit normal transcription rates and regulated gene expression in the native chromatin landscape of *T. kodakarensis* in optimal conditions.

## Modeling the impacts of histone variants on archaeal chromatin structure

HTkA$^{WT}$, like most canonical archaeal histones, is just 67 amino acids long and thus substitution of even a single residue can significantly alter the

charge and DNA interactions of the histone dimer (the minimal protein unit capable of stable DNA binding). The HTkA$^{E19K}$ and HTkA$^{G52K}$ variants have obvious impacts on transcription elongation rates and pausing patterns that are easily explained by additional hydrogen bonding between positively charged surface residues of the histone dimer with the phosphate backbone of the bound and wrapped DNA. The rationale for the impacts of other HTkA$^{variants}$ on elongation rates and pausing patterns necessitates that we model the impacts of amino acid substitutions on the totality of histone-histone and histone-DNA interactions within the extended archaeal histone-based chromatin superstructure. We calculated the impacts of select amino acid substitutions within the extended, histone-based chromatin structure composed of three histone dimers in complex with ~90 bps of DNA (PDB:5T5K)[4] in PyRosetta-4[59,60] (Fig. 6, Supplementary Data 2).

The HTkA$^{T55L}$ and HTkA$^{R20S}$ variants were known to reduce DNA affinity[1,29]. An important interaction between these residues - from one monomer to the next within the histone dimer, known as the R-T pair - was first predicted, then demonstrated through structural studies[29] (Fig. 4a). Interactions of T55 and R20 position R20 for stabilizing interactions with several nucleotides of bound DNA (Fig. 6a; green nucleotides). Substitution of R20 with serine eliminates interactions with four nucleotides of the bound DNA (Fig. 6b; gray nucleotides). The increased bulk and hydrophobicity of the T55L substitution disrupts the important salt bridge with R20[29], compromising the alignment of R20 for idealized DNA interactions and considerably increases the Lennard-Jones repulsive term (in kcal/mol)[60,61] between T55L, R20, and a neighboring nucleotide (Fig. 6c; red nucleotide).

Elongation rate reduction on templates bound by HTkA$^{G17D}$ was predicted to result, at least in part, from the impact of a bulky substitution resulting in the separation of adjacent gyres of the superhelix[4,42]. Introduction of an aspartic acid at position 17 results in steric hindrance with the preceding residue (A16) that is a part of the conserved AGA motif necessary for the tightly packed L1-L1 interface[4,5]. De-compacting archaeal-histone based chromatin might be predicted to facilitate transcription elongation, not hinder such, and modeling revealed additional impactful changes to histone-based chromatin that likely explain the appreciable challenges HTkA$^{G17D}$-based chromatin presents to TECs. Accommodating the additional bulk of an aspartic acid side chain results in clashes with the prior residue (A16), increasing the Lennard-Jones repulsive term (kcal/mol)[60,61] between D17 and A16 (Fig. 6d, e). Substitution of G17D not only results in clashes with its direct neighbor but, dramatically increases the Lennard-Jones repulsive term within the L1-L1 pocket between a native aspartic acid at position 15 (Fig. 6d, e) and several other residues within this pocket (Supplementary Data 2). Additionally, the backbone beta carbon of G17D makes a hydrogen bond with the oxygen in the carboxyl backbone group of D15 (Fig. 6f). The impacts to the local environment between D17 and A16, in combination with the increase in repulsive energy and newly acquired hydrogen bonds within the L1-L1 interface, likely alter not only the entire dimer structure but the stacking interactions of separate gyres of DNA, which would present a substantial barrier to transcription elongation (Fig. 6d–f).

Previous experimental evidence suggests that the HTkA$^{G17D}$ variant is unable to form a stable, extended superhelix from genomic chromatin

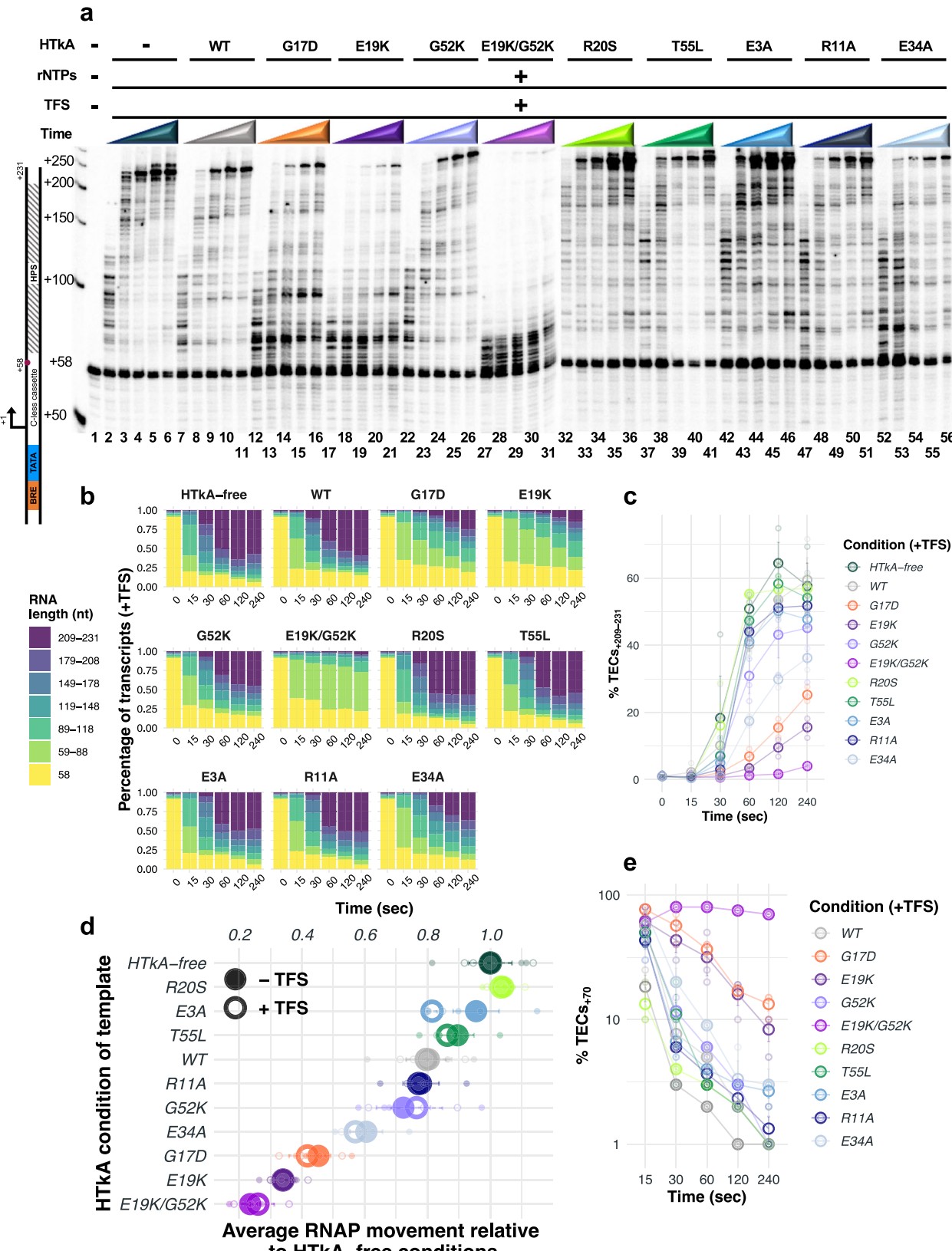

isolated from *T. kodakarensis*[4]. To observe any chromatin structure changes from the HTkA[G17D] variant in our in vitro assays, we incubated HTkA[WT], HTkA[G17D], and the HTkA[E19K/G52K] variant with the DNA template (CT3 – 298 bp) and collected aliquots over time following the addition of Micrococcal Nuclease (MNase, 15 U) (Supplementary Fig. 6). As expected,

the HTkA[E19K/G52K] variant protects most of the DNA template and stays relatively intact for extended timepoints. The HTkA[G17D] variant does indeed disrupt the normal MNase digestion patterns noted in the HTkA[WT] conditions (Supplementary Fig. 6), but in contrast to previous results[4], the HTkA[G17D] variant protects greater sizes of DNA over time than in the WT

**Fig. 5 | TFS increases productive elongation in native and most variant archaeal histone-based chromatin environments. a** Continued RNA synthesis from TECs$_{+58}$ (lane 1) was monitored identically to Fig. 1c, with the addition of TFS following transcription restart upon rNTP addition. Nascent transcript length was detected by collecting 15-, 30-, 60-, 120-, and 240-s aliquots. Radiolabeled ssDNA makers provide size standards. A representative gel image is shown; $n = 3$. **b** Stacked bar plots quantify the changing percentage of differing transcript lengths (y-axis), divided into seven distinct bins as detailed in the key (left) over time (x-axis), during elongation. The addition of TFS in vitro modestly impacts the percentage of RNA products within each bin. **c** Addition of TFS has a positive impact on the abundance of full-length RNA transcripts in all conditions but is unable to accelerate TECs through HTkA$^{E19K}$-, HTkA$^{E19K/G52K}$-, and HTkA$^{G17D}$-based chromatin landscapes to match the RNA transcript abundance in HTkA$^{WT}$ with no TFS conditions (Fig. 2b). **d** Relative RNAP elongation rates in the absence and presence of TFS are plotted with respect to the rate of synthesis (nt/s) on templates lacking histone proteins. Closed circles and open circles, respectively, detail the average elongation rates in the absence and presence of TFS addition. Error bars represent the SE from -TFS, $n = 4$ and +TFS, $n = 3$ experiments. **e** Addition of TFS does not effectively alter the +70-prominent pause, implying TECs are not backtracked due to chromatin-based impediments to translocation.

conditions, with protection sustained for longer in ~30 bp increments. From our modeling, there is an increase in the Lennard-Jones repulsive term within the L1-L1 interface (Fig. 6d, e); however, there is also an increase in the Lennard-Jones attractive term within this pocket (Supplementary Data 2). Together, these results (Fig. 6e, f, Supplementary Data 2) could account for our in vitro outcomes (Figs. 1c and 5a) and the larger MNase digestion patterns (Supplementary Fig. 6) in the HTkA$^{G17D}$ conditions when compared to the HTkA$^{WT}$ conditions.

## Substitutions to specific histone residues in an archaeal-eukaryotic histone ancestor likely assisted the evolution of chromatin-remodeling systems and extensive PTMs in eukaryotes

Increasing evidence supports eukaryogenesis as an evolved symbiosis between a bacterium (future mitochondria) and an archaeon (future nucleus) with substantial horizontal gene transfer[18–23]. While the engulfing archaeal cell provided information processing mechanisms and proteins – including the core histone fold – the bacterial cell contributed energy and lipids[18,19,21]. Given an expanding genome, gene transfers, and gene duplication events, it is highly probable that variations to key histone residues provided a pathway to the complex regulatory mechanisms we see in all eukaryotes today. Previous studies have found that there are several residues within HMfB that are structural homologs to residues within H3 and H4 essential for tetramerization in both the HMfB tetramer and the (H3/H4)$_2$ tetramer[62]. Additionally, natural archaeal histone variants with an extended α1-L1 region that consist of four additional residues in the C-terminus encode a lysine at the exact position of H3K79 (which is also an inserted sequence) that is a target for PTMs[1].

Therefore, it is reasonable that once E19K or G52K histone variants emerged, transcription elongation rates would have been significantly compromised. The impact of lysine residue substitutions to archaeal histones is nearly identical to the impacts to transcription when DNA is bound by the unmodified eukaryotic (H3/H4)$_2$ tetramer in vitro, where elongation was essentially blocked, even upon the addition of elongation factors (TFIIF and TFIIS)[63]. The reduction in transcription rates would likely favor evolution of systems to permit PTMs to selective residues of eukaryotic histone proteins to counter the impacts of increased histone-DNA complex stability due to favorable interactions between phosphates and positively charged lysine residues. The positions of HTkA$^{E19K}$ and HTkA$^{G52K}$ in archaeal chromatin nearly identically match the positions of H4K79, H4K77, H4K44, and H3K115 in eukaryotic chromatin that are common targets of PTMs to change local chromatin structures in eukaryotes: H3K115ac facilitates nucleosome repositioning[64], H4K44ac favors open chromatin configurations[65], and H4K77ac and H4K79ac facilitate DNA unwrapping and transcription factor binding[66] (Fig. 7).

## Discussion

The structure of archaeal histone-based chromatin plays a critical role in cellular viability, gene expression, and in vitro transcription activities. Our in vitro results detail the effects that specific histone residue substitutions have on the processivity of TECs in a chromatin environment. Evident changes in RNA synthesis rates resultant from altering the chromatin landscape with just a single amino acid substitution in single histone-isoform archaeal chromatin are profound and demonstrate that archaeal-histone based chromatin structures are a major regulatory force for gene expression in *T. kodakarensis*. HTkA$^{WT}$-based chromatin is a modest impediment to transcription elongation, reducing transcription rates ~20%. In comparison, HTkB-based chromatin inhibited TEC progression by ~80% in vitro and TFS addition increased elongation rates ~4-fold[6]. While these two proteins have ~85% homology, there are 11 amino acids differing between the two (none of which were studied here), resulting in varied isoelectric points that can have dramatic effects on protein-DNA interactions[29,67].

Chromatin assembled from HTkA$^{variants}$ with known reductions in DNA affinity does not present a significant barrier to transcription elongation. On HTkA$^{WT}$ or reduced-histone affinity chromatin, pausing of transcription is sporadic, short-lived, and unlikely to provide considerable regulatory potential. In contrast, substitutions that impact histone-histone and histone-DNA interactions to alter or strengthen histone-DNA contacts substantially reduce elongation rates and generate long-lived pause sites that dramatically impact elongation kinetics. As transcription and translation are coupled in archaea[68], changes in chromatin structure that impact elongation rates are likely to subsequently modify translation rates and influence the control afforded by transcription termination mechanisms dependent on access to the TEC via the nascent transcript[69–71].

The archaeal RNAP can independently traverse all the single-histone isoform chromatin landscapes we generated, albeit at different rates. Archaeal histone-based chromatin structures elicit a series of transcription pauses, particularly when the TEC first encounters DNA-bound by histones, that provide regulatory potential. These pauses are largely not resultant from backtracking of RNAP upon collisions with downstream protein barriers as might be expected. The minor impacts upon addition of TFS, the well-conserved elongation factor that stimulates endonucleolytic cleavage of the nascent transcript by RNAP, in vitro are now matched by evidence that TFS is not required for viability of *T. kodakarensis*. Given relatively minor phenotypic effects due to deletion of TFS, and failure to

## Table 1 | Residue substitutions alter TECs$_{+58}$ half-life

| Half-life at + 58 relative to HTkA$^{WT}$ (−TFS) | | |
|---|---|---|
| | t$_{1/2}$ (sec) | |
| Condition of template | −TFS ($n = 4$) | +TFS ($n = 3$) |
| *HTkA-free* | 0.28 | 0.19 |
| *WT* | 1.00 | 0.54 |
| *G17D* | 0.65 | 0.40 |
| *E19K* | 0.84 | 0.59 |
| *G52K* | 0.77 | 0.38 |
| *E19K/G52K* | 0.44 | 0.41 |
| *R20S* | 0.30 | 0.18 |
| *T55L* | 0.28 | 0.17 |
| *E3A* | 0.27 | 0.19 |
| *R11A* | 0.27 | 0.17 |
| *E34A* | 0.56 | 0.29 |

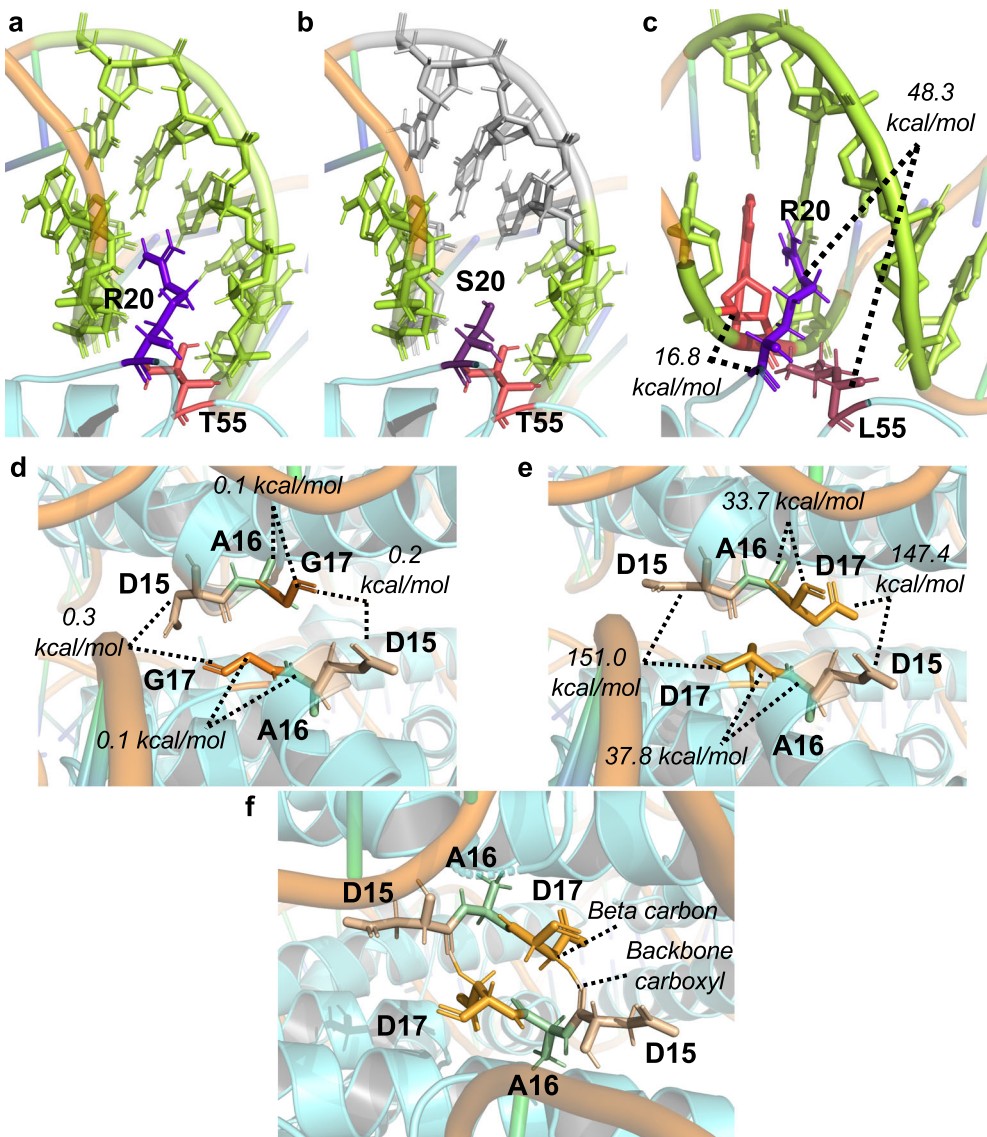

**Fig. 6 | Molecular modeling reveals the changes in fundamental molecular interactions of archaeal histone-based chromatin. a** Chromatin formed with HTkA[WT] permits interactions between T55 (pink) and R20 (purple) that position R20 for electrostatic interactions with surrounding nucleotides (green) of the wrapped DNA. **b** Substitution R20S results in lost interactions with four nucleotides (gray) that normally stabilize the archaeal histone-based chromatin landscape. **c** Substitution T55L crowds the histone-dimer interface, driving rearrangements that increase the Lennard-Jones repulsive term (kcal/mol) between T55L (maroon), R20 (purple) and an adjacent nucleotide (red). **d** The absence of a side chain on residue 17 (G17) eliminates conflicts with neighboring residues and permits tight association of the gyres of archaeal histone-based chromatin. **e** Substitution G17D drives rearrangements of A16 and D15 (from an adjacent monomer at the L1-L1 interface) that increases the Lennard-Jones repulsive energy, which impede tight gyre association and impact the 3D structure of archaeal histone-based chromatin. **f** G17D creates new hydrogen bonds with a native aspartic acid at position 15 on an adjacent gyre at the L1-L1 interface.

identify any additional archaeal-encoded factors that rescue backtracked TECs through stimulating cleavage of the nascent transcript, backtracking of TECs in vivo due to chromatin is not anticipated to be a critical regulatory property of archaeal-histone based chromatin in optimal conditions. It is likely that Spt4/5 is sufficient to accelerate TECs through chromatin-barriers in vivo. However, future evaluation of strains lacking TFS under stress conditions are imperative in understanding the broader implications of TFS not only in an altered chromatin environment, but the potential novel regulatory effects TFS may have at specific sequence elements within the *T. kodakarensis* genome[72,73].

Many of the HTkA[variants] investigated here in vitro are known to have profound effects on in vivo gene expression[4,5,29,42]. All attempts to introduce HTkA[variants] that appreciably decrease DNA-binding (HTkA[T55L] and HTkA[R20S]) into *T. kodakarensis* strains that lack HTkB were unsuccessful[4], implying that chromatinization of the genome from at least one histone protein is required for viability. In contrast, single, histone-isoform encoding strains of *T. kodakarensis* with HTkA[E19K], HTkA[G52K], or HTkA[E19K/G52K] were viable but displayed diminished growth and fitness. Compromising the extended, 3D super-helical structure of archaeal chromatin results in fitness challenges and substantial changes to the steady-state transcriptome in *T. kodakarensis*[42]. The significant impact of HTkA[G17D]-based chromatin on pausing and elongation rates implies that the normally tightly compacted, extended archaeal histone-based chromatin structure facilitates elongation, contrary to histone-based chromatin structures with modified, extended polymers[4,54] (Supplementary Fig. 6).

The impact of 3D structure for elongation rates and gene regulation is likely to have impacts beyond archaeal systems. Eukaryotic telomeric chromatin forms a columnar structure[33] much akin to the extended, super-helical structure of archaeal chromatin[7], suggesting that eukaryotic chromatin may retain additional features that match the primordial archaeal

**Fig. 7 | Archaeal- and eukaryotic-histone tetrasomes coordinate wrapped DNA nearly identically. a** Key residue substitutions that increase DNA affinity within the archaeal tetrasome closely match the positions of well-defined residues within the eukaryotic (H3/H4)$_2$ tetrasome (**b**) that can be post-translationally modified to regulate gene expression.

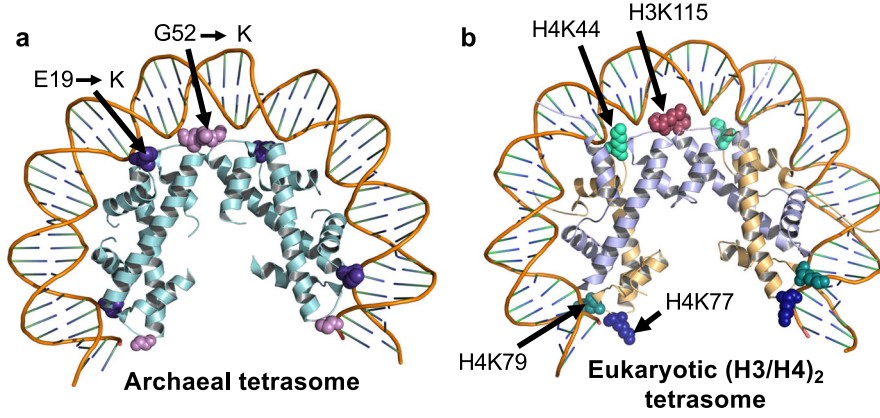

chromatin systems and regulatory strategies. Cryo-EM of archaeal-histones in complex with DNA often reveal the canonical super-helical extended structure, but also reveal a minor subpopulation of complexes with a 90° bend, altering the stacking of individual gyres of DNA into a lid-like structure[7]. The ability of archaeal chromatin to dynamically breathe into open and closed conformations, like that of telomeric chromatin, could be a key feature in regulatory mechanisms and gene expression.

In contrast to *T. kodakarensis* and several other archaeal clades, some halophilic archaea express histones at moderate levels, limiting their ability to act as general nucleoid associated proteins and subsequently more like site-specific transcription factors, and archaeal chromatin is unlikely histone-based[19,40,71]. In the many archaeal clades wherein histones are present in sufficient quantities to bind and wrap much or all of the genome(s)[6], and thus more closely mimic eukaryotic genomes, expression of different histone isoforms could greatly alter transcription processes and genomic architectures. Many archaea encode for more than one histone protein, each with predicted and known differences in DNA binding capacity, tetramer formation, and stability[19,41,54]. Histone exchanges are important in eukaryotes and provide crucial regulatory mechanisms at specific stages of development (e.g., exchange of H3.3 and H3.1 that differ in just 5 amino acid residues)[74,75]. Archaeal histone isoform exchange would provide archaeal organisms that rely on histone proteins for DNA compaction the opportunity to reliably express critical genes under specific circumstances[40,41,76]. For example, even closely related HMfA and HMfB, which share ~85% homology, differ in DNA binding affinity and in total abundance throughout the growth phase, suggesting that each isoform has a unique function[41,62,67,76]. It will thus be critical to continue to evaluate the impacts of different archaeal histone isoforms on transcription processes and cellular fitness, as these ancient DNA-binding proteins provide a platform for complex regulatory mechanisms that have come to dominate much of eukaryotic gene expression and regulation.

## Materials and methods
### Expression constructs for HTkA and site-directed mutagenesis
Mutagenesis was performed on the plasmid pTS600 which encodes TK1413 with the QuikChange II XL kit (Agilent Technologies). Codons were exchanged to those that encode for variant residues G17D, E19K, G52K, E19K/G52K, R20S, T55L, E3A, R11A, and E34A.

### Protein purifications
RNAP (RpoL-HA-His$_6$), TFB, and TBP were purified as described[58,77]. HTkA$^{WT}$, HTkA$^{G17D}$, HTkA$^{E19K}$, HTkA$^{G52K}$, HTkA$^{E19K/G52K}$, HTkA$^{R20S}$, HTkA$^{T55L}$, HTkA$^{E3A}$, HTkA$^{R11A}$, and HTkA$^{E34A}$ were expressed and purified from Rosetta 2 (DE3) cells (Millipore Sigma) cultured in Luria-Bertani broth supplemented with 270 μM ampicillin and 77 μM chloramphenicol. Expression was induced by addition of 0.5 mM isopropyl β-D-1-thiogalactopyranoside and cultures were grown for 3 h at 37 °C with shaking (200 rpm). Biomass was harvested via centrifugation and lysed via sonication in 50 mM Tris-HCl pH 8.3 and 100 mM NaCl (3 mL/g pellet).

The cell lysate was centrifuged at 30,000 × g for 30 min at 4 °C, supernatant was removed, and then spun again at 67,600 × g for 30 min at 4 °C. The supernatant containing the histones was treated with 20 μg/mL DNase I and 5 mM MgCl$_2$ at 37 °C for 2 h and then heat-treated at 85 °C for 1 h. The heat-treated lysate was clarified by centrifugation 67,600 × g for 30 min at 4 °C. The heat-treated clarified cell lysate was adjusted to a pH 6.0 and loaded onto a 5-mL HiTrap Heparin column (Cytiva) using an AKTA Pure FPLC system (GE Healthcare). Proteins were eluted over a 60-mL gradient to 50 mM Tris-HCl pH 7.0 and 1 M NaCl. Fractions containing histones were identified by sodium dodecyl sulfate (SDS)-PAGE and pooled. The pooled fractions were then concentrated to ~2 mL using Vivaspin 20, 3 kDa MWCO centrifugal concentrators (Sartorius). The concentrated pooled material was loaded over a HiPrep 16/60 Sephacryl S-100 HR equilibrated with 3 M NaCl, 50 mM Tris-HCl pH 6.8, and 5 mM 2-mercaptoethanol. Proteins were collected over a 130 mL elution in the same buffer. Fractions containing histones were identified by SDS-PAGE and pooled. The pooled fractions were dialyzed into storage buffer (50 mM NaCl, 20 mM Tris-HCl pH 7, 50% glycerol). Dialyzed proteins were quantified using a Qubit Protein Assay (Invitrogen).

### Tris-Tricine SDS-PAGE and Western blot analysis
Purified histones (1 μg) were resolved on 16.5% Mini-PROTEAN® Tris-Tricine gels and stained with Coomassie brilliant blue (Supplementary Fig. 2b) or detected via Western blot using polyclonal anti-HTkA antibodies as described (Supplementary Fig. 2c)[6].

### In vitro transcription
The DNA template used in transcription assays was generated via PCR and gel purified as described[77,78]. Assembly of preinitiation complexes (PICs) and elongation via NTP deprivation was completed as described, replacing Tris-HCl pH 8.0 with Tris-HCl pH 7.0[46,77,78]. Stalled TECs$_{+58}$ (10 nM) were chilled to 4 °C and then captured via RpoL-His$_6$ affinity with HisPur™ Ni-NTA Magnetic Beads (Thermo Scientific). TECs$_{+58}$ were washed (x 3) in 180 μL WB (20 mM Tris-HCl pH 7.0, 1 mM EDTA, 0.5 M KCl, 10 μM ATP, GTP, UTP, 4 mM MgCl$_2$, 0.1 mg/mL BSA, 0.2% glycerol) then resuspended in 10 mM Tris-HCl pH 7.0, 125 mM KCl, 5 mM MgCl$_2$, 1 mM DTT, containing 10 μM each of ATP, GTP, and UTP. The resuspended TECs$_{+58}$ were incubated with 3.5 μg of HTkA$^{WT}$ or HTkA$^{variant}$ or histone storage buffer for HTkA-free for 20 min on ice. Elongation was reinitiated at 85 °C with the addition of 25 μM ATP, GTP, CTP, UTP (and ~9 μM TFS in the +TFS conditions), removing aliquots after 15, 30, 60, 120, and 240 s (with the addition of 480 and 960 s timepoints in the extended reactions) directly to 1.2X Stop Buffer (0.6 M Tris-HCl pH 8.0, 12 mM EDTA). Radiolabeled transcripts were recovered by addition of 15 μg of GlycoBlue™ coprecipitant (Invitrogen) following an equal volume phenol/chloroform/isoamyl alcohol (25:24:1, v-v:v) extraction, and precipitation of the aqueous phase with 2.6 volumes 100% ethanol. Precipitated transcripts were resuspended in 95% formamide, 0.1% bromophenol blue, 0.1% xylene cyanol, 20 mM EDTA, heated to 95 °C for 1 min, rapidly chilled on ice, loaded, and resolved

in a 12% polyacrylamide/8 M urea, 1X TBE denaturing gel. Radiolabeled RNA was detected using Typhoon™ FLA 9500 (GE Healthcare). Gel images were analyzed using ImageQuant TL 8.2 software (Cytiva).

### In vitro transcription exchange assay

Chromatinized, stalled TECs$_{+58}$ were assembled as above with HTkA$^{E19K/G52K}$. Following elongation restart as described above, 15- and 120-s aliquots were removed, and the reaction was stopped as above. The remaining volume was split into three separate reactions at 85 °C before 16.5 µg of HTkA$^{WT}$, HTkA$^{E19K/G52K}$, or the equivalent volume of storage buffer, respectively, was added, followed by 1-, 5-, and 10-min aliquots directly to 1.2X Stop Buffer and processed as above.

### Stacked bar plots of nascent transcript length

The products of in vitro transcription resolved in each lane were parsed into seven bins based on RNA lengths determined using a linear regression of pixel positions of known molecular weight standards from the 1D gel analysis in ImageQuant™ TL 8.2.0.0 (Cytiva). The mean percentage was calculated and used in ggplot2 to create stacked bar plots in both −/+TFS conditions (RStudio 2022.07.2 + 576 for macOS).

### Average RNAP elongation rate calculations

The percentage of RNA transcripts parsed into the seven bins from the stacked bar plots was used to determine the average RNAP elongation rate (nt/s). The product from the percentage of transcripts and the theoretical RNA length (middle value within each bin) of each bin was used to determine the average transcript length at each timepoint within each environment. The increase in the average RNA length was determined by taking the difference between the average length of RNA and the starting point (+58). The average RNAP elongation rate (nt/s) was calculated by taking the mean of the increase in the RNA average length divided by each timepoint.

### Pause half-life calculations

To calculate the rate constant (k) at position +58 we determined the average percentage of complexes in bin 1 (+58 nt) from the 1D analysis in ImageQuant™ TL 8.2.0.0 (Cytiva) from the stacked bar plots and the formula: $C2 = C1e^{-k(t2-t1)}$; where C1 = the average percentage of complexes in bin 1 at 15 s, C2 = the average percentage of complexes in bin 1 at 240 s, t1 = 15 s, t2 = 240 s. The rate constant (k) was then used in the formula: $t_{1/2} = \ln2/k$ to determine the average half-life of complexes at position 58 in -/+ TFS conditions.

### Strain construction and growth conditions

*Thermococcus kodakarensis* strains were constructed as described[42,58]. Strain RLV2 was constructed via markerless deletion of TK0533 (TFS). Deletion was confirmed by PCR amplification with primers flanking TK0533 and whole genome sequencing (WGS) on our in-house MinION, which contains the sequencing software, MinKNOW. MinKNOW does a post-run analysis that utilizes Guppy for base-calling, minimap2 for alignment with the reference genome, and medaka to call SNPs/indels, which was then visualized using IGV genome browser (2.16.1). Cultures were grown as described[42].

### Molecular modeling

To determine the predicted impacts of select HTkA$^{variants}$ on archaeal-histone based chromatin structures, we loaded the 5T5K PDB structure into PyRosetta-4[59]. In addition, we utilized the PyMOL generate symmetry mates function to stack the 5T5K PDB structure and create a new PDB file to observe the energy within the L1-L1 energy pocket. The PDB structure was cleaned using the cleanATOM function and then relaxed using several PyRosetta functions (FastRelax and MoveMap) with backbone, sidechain, and start coordinate constraints[79,80]. The selected residues were then mutated utilizing the mutate_residue module. To determine the local energy contribution of each residue that was substituted to compare to the WT energy at that position, we utilized the ability of PyRosetta to store the total, residue, residue-pair, and residue neighbor energy information. The core

scoring energy function was used on each variant to discern the contributing energy terms associated with each substitution at the given position. To observe the most considerable energy contributions, we compared the WT and variant ScoreType that calculated the energy score of residue pairs at a given residue (pyrosetta.toolbox.atom_pair_energy.print_residue_pair_-energies(*res, pose, score_function, score_type, threshold = 0*))[59,60,79,80]. The PyRosetta_HTkA_modeling.html file (Supplementary Data 2) provides a step-by-step visualization of the input and output to obtain our results.

### MNase digestion of the in vitro DNA template

Digestion of the DNA (CT3 – 298 bp) was completed as described[4], with a few alterations. 50 µL reactions containing 134 nM of CT3 template, 100 mM KCl, and 50 mM Tris-HCl, pH 7.0 were incubated with either storage buffer (50 mM NaCl, 20 mM Tris-HCl pH 7.0, 50% glycerol) or 20 µg of HTkA$^{WT}$, HTkA$^{G17D}$, or HTkA$^{E19K/G52K}$ for 15 min at room temperature and then 15 min at 4 °C. Reactions were diluted to 100 µL with 5 mM CaCl$_2$, 0.1 mg/mL BSA, 15 U Micrococcal Nuclease (MNase, Thermo Scientific) and 50 mM Tris-HCl, pH 7.0, where control reactions (0 min timepoints) lacked MNase. 100 µL reactions were incubated for 0, 3, 6, and 12 min at 37 °C and reactions stopped with 25 µL of 0.5 M EDTA, pH 8.0. 40 µg of Proteinase K was added to each reaction, incubated for 30 min at 55 °C, and then purified with the Monarch PCR & DNA Cleanup Kit (NEB) utilizing the Oligonucleotide Cleanup Protocol. The purified DNA fragments were visualized on a 10% Criterion TBE-Urea Polyacrylamide Gel (BioRad).

### Reporting summary

Further information on research design is available in the Nature Portfolio Reporting Summary linked to this article.

## Data availability

The mapped WGS reads along with the reference file are publicly available through links to BioProject accession number PRJNA996631 in the NCBI BioProject database (https://www.ncbi.nlm.nih.gov/bioproject/). The source data for Figs. 2c and 5d can be found in Supplementary Data 1. All raw images used for data analysis and as representative images are available upon request and included as Supplementary Fig. 7 (a – l).

## Code availability

The code for the Pyrosetta analysis to determine the free energy of the wildtype and mutant histone-based chromatin structures can be found in Supplementary Data 2 and is available, along with the R scripts for data analysis, on our GitHub repository: https://github.com/tjsantangelo/Commun-Biol-archaeal-histones, https://doi.org/10.5281/zenodo.10553889.

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

## Acknowledgements

We thank members of the Santangelo lab for assistance editing the manuscript. This work was supported by the USA National Institutes of Health, General Medical Sciences, grants GM143963 and GM100329, to T.J.S. BRW was supported, in part, by the National Science Foundation NRT Grant No. 1450032.

## Author contributions

B.R.W. purified archaeal histones, performed in vitro transcription assays, built structural models, analyzed data, prepared figures, and assisted with manuscript preparation. R.L.V. purified archaeal histones, and R.L.V. and B.W.B. generated *T. kodakarensis* strains. T.J.S. conceived and directed the project, analyzed data, and wrote the manuscript.

## Competing interests

The authors declare no competing interests.
