## [Peer review file · Communications Biology]

Reviewers' comments:

Reviewer #1 (Remarks to the Author):

In the paper called "Archaeal histone-based chromatin structures regulate transcription elongation rates" by Breanna R. Wenck, Robert L. Vickerman, Brett W. Burkhart, and Thomas J. Santangelo, transcription elongation is monitored in vitro from naked DNA, and DNA bound by a variety of histone variants. The histone protein under study is one of two histones of *Thermococcus kodakarensis*, HTkA. The different mutant variants analyzed have different DNA- or histone binding affinities. In addition, the role of the cleavage stimulatory factor TFS is studied. Results show that histone-barriers slow down and temporarily halt elongation (thereby this can be a way of regulating gene expression) and that the role of TFS in resolving this pause is minimal. My overall impression of this paper is positive. The idea behind this study is interesting, findings are relevant in the field, the experiments are technically sound and they provide sufficient depth to the study (e.g. the supplementary in vitro transcription experiments for E19K/G52K performed for longer durations or upon dilution of the mutant histone pool). The paper is well-written and the M&M section is described in sufficient detail to allow reproducibility. The study provides a good starting point for future follow-up experiments, e.g. in an in vivo setting. My major remark is related to the numerous figures which are referred to at the same time and in the complete results-section. This makes that it is often very difficult to follow and to know what one should "see" exactly on the different figures. I wonder whether a reorganization of the different figures might help in resolving this issue. Besides this, I only have very little comments.

Major remark:

1) The results section is quite difficult to follow with references to multiple figures in the main text and supplementary material at the same moment (although I believe each individual figure is meaningful). The fact that Figures 1-4 are referred to during the whole results-section further complicates this. I would strongly advise to tweak the organization of the manuscript by rethinking the grouping of the figures in the main text. This is especially relevant for the last section regarding TFS. Would grouping all TFS-related data into one figure be an option (because e.g. "Figs. 2 & 3; Supplementary Figs. 6, 7 & 8; Table 1; Supplementary Data 1" (L295) is a bit cumbersome)? Furthermore, perhaps certain supplementary figures could be added as a subfigure to a main text figure, because I feel like they are really meaningful for the interpretation of the results described (e.g. Supplementary Fig. 1a, Supplementary Fig. 5, Supplementary Fig. 7, Supplementary Fig. 8,...).

Minor comments:

1) I am wondering whether the individual functions of the two histone proteins (HTkA and HTkB) are known (perhaps related to stress response?)? Why was HTkA selected as the histone protein of interest in this study? And to which extent do you think the major conclusions will be similar for the other histone variant, HTkB, or the heterodimer HTkA-HTkB? It might be interesting to elaborate on this in the discussion-section.

2) L 143 "HTkA normally functions at 85-95°C".

It would be interesting to put this in context of the optimal growth temperature of *T. kodakarensis*.

3) L 152-154 "To ensure addition of archaeal histones and resultant chromatin structures did not impede transcription initiation, stalled TECs were first formed on histone-free DNA via initiation at a C-less cassette with only ATP, UTP, and GTP."

I am not familiar with this approach and do not understand how a C-less cassette exactly leads to a stalled TEC. A brief explanation in the text would be helpful, or alternatively, a reference to a previous research paper doing so.

4) L 204-205: "Introduction of histone variants R20S (Fig. 1b, lanes 32-36) and T55L (Fig. 205 1b, lanes 37-41) decrease DNA affinity and result in minimal elongation conflicts". + L 208-210 "In stark contrast, introduction of histone variants E19K (Fig. 1b, lanes 17-21), G52K (Fig. 1b, lanes 22-26), or a variant with both E19K/G52K (Fig. 1b, lanes 27-31) that significantly increase DNA affinity result in..."

Is the affected DNA affinity something that is hypothesized based on previous studies (if so, a

reference should be added)? Or is this something that could be observed/derived from the data shown?

5) Supplementary Fig 7: "TECs remaining after 2 minutes were split into thirds, and either storage buffer, HTkAWT, or additional HTkAE19K/G52K was added, before allowing additional time for continued RNA synthesis."

The amount of additional time should be specified in the legend (up to 12 min?).

6) L 240-249: A conclusion-sentence seems to be missing for this dilution experiment.

7) L 334-336 "Despite the positive effects of TFS on backtracked TECs traversing some chromatin landscapes in vitro, the absence of TFS does not result in any notable growth defects (Fig. 11b)". I would advise to be a bit more careful with this statement, because RLV2 seems to grow (a bit) more slowly compared to the WT strain, based on the figure. Also note here that the reference should be Supplementary Fig 11b.

8) References to the correct figures and lanes within figures should be verified. Eg:

o L 143-147: "HTkA normally functions at 85-95°C and preparations of recombinant HTkA often retained dimeric-interactions even after extensive heating and SDS-PAGE (Supplementary Fig. 1a). Western blotting with anti-HTkA antibodies confirm that the higher order complexes resolved in SDS-PAGE are oligomerized HTkA complexes (Supplementary Fig. 1b).".

I believe it should be referred to Supplementary Fig 1b and 1c, respectively, instead.

o L 169 "Elongation on histone-free templates (Fig. 1b, lanes 2-7)".

Wouldn't this be lanes 2-6?

9) The consistency in layout of the different figures could still be improved (e.g. font, colors,...).

10) L 111: abbreviation of TEC is only introduced at L 130.

11) L 269: typo dimeraztion

Reviewer #2 (Remarks to the Author):

In the manuscript titled "Archaeal histone-based chromatin structures regulate transcription elongation rates," the authors investigated the influence of archaeal histone variants on transcription elongation within a chromatin context using an in vitro transcription system. They introduced various amino acid substitutions to modify histone-DNA affinity or the degree of histone oligomerization and rigorously analyzed the transcription elongation rate through chromatin. The findings revealed that amino acid substitutions which enhance histone-DNA affinity or alter histone oligomerization, strongly inhibit transcription elongation. The results provide detailed understanding on how RNAP moves through a chromatin and shed light on the evolution of histone-based chromatin system. The study is well-executed, employing appropriate methods and presenting data clearly. It will be of interest to researchers studying archaeal chromatin structure and being interested in the evolution of chromatin regulation system.

Major comment:

1. The argument regarding the effect of the amino acid substitution G17D is based on the assumption that a 3D chromatin structure is formed. However, it remains unclear whether the 3D chromatin structure is indeed established on the "tandem 60 bp SELEX-derived histone positioning sequence." (line 160) of the template DNA used in the in vitro transcription experiment. To enhance the clarity and rigor of the study, please provide an explanation in the text that establishes the existence of the 3D chromatin structure within the experimental context.

Minor comments:

2. In the abstract and on page 5 of the introduction, the abbreviation "TFS" is used without prior explanation. Please spell out the full form of "TFS" upon its first use.

3. In the figure legend on page 35, lines 844-849, there seems to be an issue with the labeling of figures. The figure legend states "Relative RNAP elongation rates in the absence (a) and presence of TFS (b)," but this description does not appear to match the actual figure. Both figures 3a and 3b appear to contain results with and without TFS. Additionally, there is no explanation provided in the legend for Figure 3b. Please revise the figure legend to accurately describe the content of the figures and provide a clear explanation for Figure 3b.

4. While Figure 3 as a whole is frequently referenced in the text, Figure 3b does not receive specific mention. Please provide the necessary context for Figure 3b in the text to help the reader understand the importance of Figure 3b.

Editors and Reviewers:

The two reviews of our original manuscript were positive and the suggestions for modifications to such resulted in an improved, clarified, and more impactful revised manuscript. Figures have been modified to improve the flow of the manuscript, inclusive of relocating some images from the supplemental to the main text (as suggested by reviewer #1). Additional experimentation (requested by reviewer #2) detailing the binding and impacts of histone-DNA interactions on chromatin structure and transcription elongation under our experimental conditions were performed and the results of such were added as additional supplemental figures. All small grammatical and typographical errors were corrected.

In addition to supplying the revised manuscript, we provide both a marked version of the original manuscript (to highlight the positions and extent of changes) and a point-by-point response to each comment from the reviewers. All authors have contributed to the changes that improve the manuscript, and all authors approve the revised manuscript. No changes to authorship were warranted.

-Tom Santangelo (on behalf of all authors)

Referee expertise:

Referee #1: Expert in archaeal gene regulation

Referee #2: Expert in archaeal chromatin structure

Reviewers' comments:

Reviewer #1 (Remarks to the Author):

In the paper called "Archaeal histone-based chromatin structures regulate transcription elongation rates" by Breanna R. Wenck, Robert L. Vickerman, Brett W. Burkhart, and Thomas J. Santangelo, transcription elongation is monitored in vitro from naked DNA, and DNA bound by a variety of histone variants. The histone protein under study is one of two histones of *Thermococcus kodakarensis*, HTkA. The different mutant variants analyzed have different DNA- or histone binding affinities. In addition, the role of the cleavage stimulatory factor TFS is studied. Results show that histone-barriers slow down and temporarily halt elongation (thereby this can be a way of regulating gene expression) and that the role of TFS in resolving this pause is minimal.

My overall impression of this paper is positive. The idea behind this study is interesting, findings are relevant in the field, the experiments are technically sound and they provide sufficient depth to the study (e.g. the supplementary in vitro transcription experiments for E19K/G52K performed for longer durations or upon dilution of the mutant histone pool). The paper is well-written and the M&M section is described in sufficient detail to allow reproducibility. The study provides a good starting point for future follow-up experiments, e.g. in an in vivo setting. My major remark is related to the numerous figures which are referred to at the same time and in the complete results-section. This makes that it is often very difficult to follow and to know what one should "see" exactly on the different figures. I wonder whether a reorganization of the different figures might help in resolving this issue. Besides this, I only have very little comments.

Major remark:

1) The results section is quite difficult to follow with references to multiple figures in the main text and supplementary material at the same moment (although I believe each individual figure is meaningful). The fact that Figures 1-4 are referred to during the whole results-section further complicates this. I would strongly advise to tweak the organization of the manuscript by rethinking the grouping of the figures in the main text. This is especially relevant for the last section regarding TFS. Would grouping all TFS-related data into one figure be an option (because e.g. "Figs. 2 & 3; Supplementary Figs. 6, 7 & 8; Table 1; Supplementary Data 1" (L295) is a bit cumbersome)? Furthermore, perhaps certain supplementary figures could be added as a subfigure to a main text figure, because I feel like they are really meaningful for the interpretation of the results described (e.g. Supplementary Fig. 1a, Supplementary Fig. 5, Supplementary Fig. 7, Supplementary Fig. 8,...).

We are happy to move some of the supplemental figures into the main figures to clarify interpretations of the data. We have altered the text and figures appropriately to make the data more accessible and easier for the reader to digest.

Minor comments:

1) I am wondering whether the individual functions of the two histone proteins (HTkA and HTkB) are known (perhaps related to stress response)? Why was HTkA selected as the histone protein of interest in this study? And to which extent do you think the major conclusions will be similar for the other histone variant, HTkB, or the heterodimer HTkA-HTkB? It might be interesting to elaborate on this in the discussion-section.

We thank the reviewer for their suggestion and have modestly expanded the discussion to address this point. At least in *T. kodakarensis*, HTkA or HTkB is individually dispensable, but at least one histone must be present for viability (Cubonova et al; 2012). Strains lacking one histone variant display only modest changes in phenotype, and older (but reliable) microarray data suggest that minimal gene expression changes occur due to deletion of either HTkA or HTkB. While it is certainly possible that unique in vivo functions of the histone variants in *T. kodakarensis* exist, given the dearth of information in support of such

(including a lack of data in support of a stress response that involves histone proteins or radical changes to chromatin architectures) we remain hesitant to expand much further as this manuscript is focused on the *in vitro* impacts of histone-DNA interactions on movement of the transcription apparatus. Given that modeling of HTkA and HTkB binding to DNA results in nearly identical superstructures (Bhattacharyya, et al., 2018; Henneman et al; 2018; Henneman et al; 2021), there is little reason to suspect that altering HTkA for HTkB here would result in significant impacts. Studies of the similarly closely related HMfA and HMfB (histones from *Methanothermobacter ferredoxigenes*) proteins suggests that archaeal histone variants are differentially expressed and have different DNA binding affinities. We address this and our stance on histone isoform exchange in our final Discussion paragraph. We agree that it will be critical to address these questions moving forward and have already designed several experiments to hopefully answer such in the future.

We partially selected HTkA as the impact of HTkB-based chromatin structures on transcription elongation was already studied (not to this extent) *in vitro* (Sanders, et al., 2019). Finally, as we have plans to investigate the impacts of select HTkA variants *in vivo*, the choice of HTkA provides a simpler platform, as deletion of HTkA results in complications for the normally facile *T. kodakarensis* genetic system.

2) L 143 "HTkA normally functions at 85-95°C".

It would be interesting to put this in context of the optimal growth temperature of *T. kodakarensis*.

We have clarified the text to clarify that the optimal growth temperature of *T. kodakarensis* is 85°C.

3) L 152-154 "To ensure addition of archaeal histones and resultant chromatin structures did not impede transcription initiation, stalled TECs were first formed on histone-free DNA via initiation at a C-less cassette with only ATP, UTP, and GTP."

I am not familiar with this approach and do not understand how a C-less cassette exactly leads to a stalled TEC. A brief explanation in the text would be helpful, or alternatively, a reference to a previous research paper doing so.

We thank the reviewer and have altered the figure legend (Fig. 1b.) in hopes that this will help make the experiment clearer to the audience. In short, as RNAP normally uses 4 substrates (ATP, GTP, CTP, and UTP), if just three of the four substrates are supplied (i.e. reactions lacking CTP), RNAP can elongate transcripts until CTP incorporation is necessary. By generating a promoter that initiates transcription but does not require CTP until after 58 nts of RNA are generated, we can stall RNAP by nucleotide deprivation by initiating on a C-less cassette in the absence of CTP.

4) L 204-205: "Introduction of histone variants R20S (Fig. 1b, lanes 32-36) and T55L (Fig. 205 1b, lanes 37-41) decrease DNA affinity and result in minimal elongation conflicts". + L 208-210 "In stark contrast, introduction of histone variants E19K (Fig. 1b, lanes 17-21), G52K (Fig. 1b, lanes 22-26), or a variant with both E19K/G52K (Fig. 1b, lanes 27-31) that significantly increase DNA affinity result in..."

Is the affected DNA affinity something that is hypothesized based on previous studies (if so, a reference should be added)? Or is this something that could be observed/derived from the data shown?

The DNA binding properties of the referenced histone variants are well studied, and we improved the manuscript through addition of references to previous work that supports the statements and conclusions in the revised manuscript.

5) Supplementary Fig 7: "TECs remaining after 2 minutes were split into thirds, and either storage buffer, HTkAWT, or additional HTkAE19K/G52K was added, before allowing additional time for continued RNA synthesis."

The amount of additional time should be specified in the legend (up to 12 min?).

We thank the reviewer for their attention to detail and have added the information to the legend (Fig. 3e).

6) L 240-249: A conclusion-sentence seems to be missing for this dilution experiment.

Rearrangements of the text have clarified the conclusions of this section.

7) L 334-336 "Despite the positive effects of TFS on backtracked TECs traversing some chromatin landscapes in vitro, the absence of TFS does not result in any notable growth defects (Fig. 11b)".

I would advise to be a bit more careful with this statement, because RLV2 seems to grow (a bit) more slowly compared to the WT strain, based on the figure. Also note here that the reference should be Supplementary Fig 11b.

We agree with the reviewer and have altered our verbiage in the text to reflect more careful language.

8) References to the correct figures and lanes within figures should be verified. Eg:

o L 143-147: "HTkA normally functions at 85-95°C and preparations of recombinant HTkA often retained dimeric-interactions even after extensive heating and SDS-PAGE (Supplementary Fig. 1a). Western blotting with anti-HTkA antibodies confirm that the higher order complexes resolved in SDS-PAGE are oligomerized HTkA complexes (Supplementary Fig. 1b)".

I believe it should be referred to Supplementary Fig 1b and 1c, respectively, instead.

o L 169 "Elongation on histone-free templates (Fig. 1b, lanes 2-7)".

Wouldn't this be lanes 2-6?

We thank the reviewer and have carefully gone through the manuscript to verify proper references to the correct figures.

9) The consistency in layout of the different figures could still be improved (e.g. font, colors,...).

We agree and have taken the time to remake several figures and changed formatting to improve the fluency.

10) L 111: abbreviation of TEC is only introduced at L 130.

We thank the reviewer and have introduced ternary elongation complexes (TECs) in the introduction.

11) L 269: typo dimeraztion

We thank the reviewer, and this has been fixed.

Reviewer #2 (Remarks to the Author):

In the manuscript titled "Archaeal histone-based chromatin structures regulate transcription elongation rates," the authors investigated the influence of archaeal histone variants on transcription elongation within a chromatin context using an in vitro transcription system. They introduced various amino acid substitutions to modify histone-DNA affinity or the degree of histone oligomerization and rigorously analyzed the transcription elongation rate through chromatin.

The findings revealed that amino acid substitutions which enhance histone-DNA affinity or alter histone oligomerization, strongly inhibit transcription elongation. The results provide detailed understanding on how RNAP moves through a chromatin and shed light on the evolution of histone-based chromatin system. The study is well-executed, employing appropriate methods and presenting data clearly. It will be of interest to researchers studying archaeal chromatin structure and being interested in the evolution of chromatin regulation system.

Major comment:

1. The argument regarding the effect of the amino acid substitution G17D is based on the assumption that a 3D chromatin structure is formed. However, it remains unclear whether the 3D chromatin structure is indeed established on the "tandem 60 bp SELEX-derived histone positioning sequence." (line 160) of the template DNA used in the in vitro transcription experiment. To enhance the clarity and rigor of the study, please provide an explanation in the text that establishes the existence of the 3D chromatin structure within the experimental context.

We thank the reviewer and agree with their assessment. To establish the 3D structure on our DNA template, we developed an MNase experiment including DNA alone,

DNA/HTkA^{WT}, DNA/HTkA^{G17D}, and DNA/HTkA^{E19K/G52K}. We added 15 U of MNase and then took 0-, 3-, 6-, and 12-minute timepoints to establish the structure with each, via the DNA digestion patterns (Supplementary Fig. 6) and discuss the results in the text. The results clearly show that under our experimental conditions that histone protein both bind the DNA template and generate 3D chromatin structures that impede the progression of the transcription apparatus.

Minor comments:

2. In the abstract and on page 5 of the introduction, the abbreviation "TFS" is used without prior explanation. Please spell out the full form of "TFS" upon its first use.

We thank the reviewer and have resolved such in both the abstract and the introduction.

3. In the figure legend on page 35, lines 844-849, there seems to be an issue with the labeling of figures. The figure legend states "Relative RNAP elongation rates in the absence (a) and presence of TFS (b)," but this description does not appear to match the actual figure. Both figures 3a and 3b appear to contain results with and without TFS. Additionally, there is no explanation provided in the legend for Figure 3b. Please revise the figure legend to accurately describe the content of the figures and provide a clear explanation for Figure 3b.

We thank the reviewer for their insight and have dramatically altered the figures/legends to be more accessible for readers. We hope the new format and explanation (Fig. 4d) sufficiently resolves the issue.

4. While Figure 3 as a whole is frequently referenced in the text, Figure 3b does not receive specific mention. Please provide the necessary context for Figure 3b in the text to help the reader understand the importance of Figure 3b.

As from the concerned stated from number 3 above, we have altered the figures and text to reflect such, and figure 3b is now Supplementary Fig. 3e.

REVIEWERS' COMMENTS:

Reviewer #2 (Remarks to the Author):

This is the second revision of the manuscript entitled "Archaeal histone-based chromatin structures regulate transcription elongation rates" by Breanna R. Wenck, Robert L. Vickerman, Brett W. Burkhart and Thomas J. Santangelo. I appreciate the author's elaborate answers and feedback on my remarks of the original submission and I am pleased with the work which was put in revising the manuscript accordingly. I support the rearrangement of the figures, which has drastically improved the flow of the manuscript and ease of interpretation of the results. I do not have any further remarks and would like to congratulate the authors with this interesting study.

Reviewer #3 (Remarks to the Author):

I appreciate that the authors have addressed the raised points very carefully. I have no further comments.